# Understanding Customer Responses to Service Failures during the COVID-19 Pandemic for Sustained Restaurant Businesses: Focusing on *Guanxi*

**Chenyu Zhang [1], Junkyu Park [1], Mark A. Bonn [2] and Meehee Cho [1,*]**

[1] College of Hotel and Tourism Management, Kyung Hee University, 26 Kyungheedae-ro, Dongdaemun-gu, Seoul 02447, Korea; ceonyu@khu.ac.kr (C.Z.); greatjkpark@khu.ac.kr (J.P.)

[2] Dedman School of Hospitality and Tourism Management, Florida State University, 288 Champions Way, Tallahassee, FL 32306-2541, USA; mbonn@dedman.fsu.edu

\* Correspondence: chom2h2@khu.ac.kr; Tel.: +82-10-3521-4434

**Abstract:** Due to the COVID-19 pandemic, restaurants worldwide, including China, have been forced to protect public health by following food safety standards and adapting to the necessary social distancing practices. Accordingly, restaurant diners who are concerned about food safety and unsure of whether it is truly safe to dine out, put more importance on the entire stages of service consumption. Restaurants must make their best efforts to minimize service failures in their service provision process and outcomes. Given that customers from different cultures are reported to evaluate service quality differently, this study was designed to investigate what actions Chinese customers who encounter service failures would take under the influence of *Guanxi*. *Guanxi* represents Chinese attitudes towards long-term individual and business relationships and ultimately involves moral obligations and mutual favors. Analyzing our structural equation model using 439 responses obtained from Chinese diners, this study determined that Chinese consumers would react differently in the service process failures and outcome failures in terms of negative word-of-mouth, direct complaints, switching intention, and revisit intention. More importantly, this study confirmed the significant moderating effects of *Guanxi* within the proposed relationships. Based on the study's findings, useful implications are provided for academics and practitioners regarding sustained restaurant businesses.

**Keywords:** service failures; Guanxi; direct complaints; negative word-of-mouth; switching intentions; revisit intentions; sustained restaurant businesses

## 1. Introduction

Looking at the dining out frequency patterns in China, a global consumer survey conducted in 2017 reported that 43% of the Chinese customers eat lunch out five times and more a week and 78% of them eat dinner out at least once a week [1]. China's food service establishments revenue in 2018 touched 4271.6 billion yuan, and the number of restaurants in 2019 reached 7,118,058 [2]. However, the coronavirus pandemic has significantly damaged the restaurant industry globally [3] and specifically in China [4]. Emerging evidence reveals that the risk of spreading COVID-19 increases in the restaurant setting, and restaurants worldwide have been forced to protect public health by adapting to necessary social distancing practices [5].

In addition, customers pay more attention to food safety and seek restaurants that perform hygiene practices including cleaning, sanitization, and disinfection [6]. Thus, restaurant guests who are unsure of whether it is truly safe to dine out, put more importance on the entire service consumption experiences from service processes to service outcomes [7]. Therefore, it has become more critical during the COVID-19 pandemic that restaurants communicate their best food and service practices clearly. In order for

customers to feel confident dining out, restaurants must make their best efforts to minimize critical flaws in their service provision process and outcomes.

However, the nature of service is not simple to define and can be highly personalized concerning customer demand and preferences [8,9]. Service quality is evaluated based on how satisfied customers are with the service outcome [10]. When customers perceive that the service provided is less than their expectations, those customers are likely to experience service failures [11,12]. In particular, service failures in the hospitality industry are inevitable because services are offered by "human beings" [13]. An early study by Hoffman et al. [14] classified service failures into a three-dimensional concept that can occur in the hospitality industry: "(1) employee responses to service delivery failures (e.g., restaurant meal defects and slow or unavailable service), (2) employee responses to customer needs and requests (e.g., failing to cook foods as requested and responding to seating preferences), and (3) unprompted and unsolicited employee actions (e.g., wrong order and mischarging)" (p. 49). This three-dimensional concept of service failures is classified into two stages involving processes and outcomes [15,16]. Process failures involve the interpersonal process of service delivery while outcome failures involve the core and impersonal service outcome [15].

As literature revealed that service failures are critical in determining a business's survival, many scholars have attempted to identify how customers react to service failures and how their actions affect the business [17]. Service failures result in customer dissatisfaction, negative word-of-mouth (WOM) behavior and customer defection [18]. In particular, the impact of WOM has become more powerful in the restaurant industry because today's consumers seek dining experiences shared by others before consumption. Consumers thus rely on WOM to make inferences about product and service quality [19]. Due to the pervasive use of online platforms to share their dining experiences, it is easier for WOM to spread more extensively and rapidly [20]. This situation leads customers who encounter service failures to spread their unpleasant dining experiences and engage in negative WOM more easily and actively [21]. This can cause a significant damage to the restaurant business, and particularly during the pandemic.

Several studies reported different customer reactions with dissatisfying service encountered in the hotel context. These include "exiting silently", "spreading a negative word-of-mouth (WOM)", "directly complaining to the service provider", and "continuing patronage despite their service failure experiences" [13,18,22]. Accordingly, this current study cross-examines why customers experiencing similar service failures show different reactions. Schutte and Ciarlante [23] proposed that an understanding of consumer behaviors would be ineffective without taking cultural differences into consideration. Following this proposition, several studies revealed that customers from different cultures perceived service quality differently [24,25] and exhibited different responses and behaviors towards the same service failure event [26,27]. Although there are existing studies on the effects of cultural differences in the cross-national comparison context, it is still important to expand the academic effort in the examination of cultural values possessed by restaurant diners. During a crisis such as the COVID-19 pandemic, a more accurate understanding of customer perceptions about what matters in a restaurant service failure context is required to identify the differences between the main triggers to switching intentions and revisit intentions. Thus, obtaining an insight into how cultural values perceived by restaurant guests involves the interpretation of service failures would be useful in developing more effective service recovery strategies.

With this information in mind, this study explored the roles that unique cultural-based traits of traditional Chinese relationships representing *Guanxi* played in the restaurant service failure setting. Generally, *Guanxi* is defined as a deep psychological commitment among Chinese people in an emphasis on mutual empathetic understanding, sharing of feelings and emotional identification rather than responsibilities or obligation [28,29]. The hospitality industry is largely human-resource-oriented and inevitably relies on various interpersonal and social interactions between customers and employees (or managers and

owners), which can be affected by their culture and society [30,31]. Those interactions between Chinese are culturally rooted based on *Guanxi*, conceptualizing informal personal connections developed based on social norms (e.g., reciprocity, mutual commitment) and long-term relationships [32]. *Guanxi* reflects in a variety of contexts (i.e., customer to business, or business to business). It is common that Chinese tend to use their personal connections developed based on *Guanxi* to attain competitive business advantages. Exchange parties connected based on *Guanxi* develop trust in their open-ended and long-term transactions and feel morally obliged to favor other parties [33]. Thus, if they do not do something beneficial for other parties, they feel like they lost face and connections [34]. *Guanxi* appears to be an important topic in academic research. As an example, several studies found that hotel managers make put a great deal of effort into developing and leveraging personal connections with important and beneficial accounts [35,36].

Applying the *Guanxi* concept to Chinese customers in the restaurant context, this study explores how customers from a different cultural background would react to restaurant service failures. It is crucial to recognize how customers who have developed personal connections (*Guanxi*) with a restaurant owner or employee may have a different viewpoint regarding service failures, leading to varying responses. In other words, when restaurant guests having high *Guanxi* encounter service failures, their responses tend to be directed towards helping the restaurant rather than expressing their anger. This may imply that *Guanxi* mitigates the negative impact of service failures on consumer negative responses. Thus, this study aimed to identify how Chinese customers respond to restaurant service failures. This study also strived to answer questions such as: Do they directly complain to the service provider? or do they spread negative WOM to others? In addition, this study examined how negative WOM, and direct complaints, were associated with their switching and revisiting intentions. More importantly, this study focused on identifying if *Guanxi* has vital moderating effects on negative WOM relationships, direct complaints, switching intentions, and revisit intentions.

## 2. Literature Review and Hypothesis Development

### 2.1. Guanxi

*Guanxi* 關係 in Chinese defines the fundamental dynamic in individualized social networks of power and is a crucial system of beliefs in Chinese culture [37,38]. *Guanxi* is explained based on tacit mutual commitments, reciprocity, and trust, and it is conceptualized as a connection in personal relationships to ensure the interest of each other [39]. *Guanxi* appears to be Chinese tradition and culture which includes various moral codes, values, and power structures built based on Confucianism; and also regulates and coordinates complex relationships among people through a strong hierarchical structure. Due to this, *Guanxi* ensures the harmony and unity of society [32,40]. In order to address the relational appropriateness between Chinese people, it is necessary to participate in *Guanxi* interactions with the commitment to signify humanize and the universe [32]. Thus, to understand why is important and how it operates in Chinese societies, one must understand personal relationships amongst human beings, representing *Guanxi*.

Likewise, *Guanxi* is very frequently developed between Chinese people with "a commonality of shared identification." Duane [41], an author of a book, "Action Selling," emphasized the powerful roles of Guanxi in relationship marketing. He stated that Guanxi is a prerequisite and basic factor to develop close relationships with consumers and it is also regarded as a debt to others for help. For instance, when others help someone, then the one has the obligation to help those people in the future as we have to repay the debt [42]. Thus, *Guanxi* can be defined as a deep psychological commitment among Chinese people in an emphasis on mutual empathetic understanding, sharing of feelings and emotional identification rather than responsibilities or obligation [43,44].

*Guanxi* is involved in both interpersonal and inter-organizational relationships in China and has been well-known to significantly influence business marketing and management, scholars and practitioners began to pay attention to identify *Guanxi* and its roles in

business [45,46]. *Guanxi* has been applied to relationship marketing strategies due to their effect on the establishment of long-term mutually beneficial relationships and trust with customers, distributors, dealers, and suppliers in order to promote the success of business transactions [47,48].

*2.2. Service Failures in the Restaurant Business*

Since service performance plays a crucial role in terms of satisfying customers specifically for the restaurant business, failures in a customer service sector become critical issues [32]. Service failures are defined as service performance perceived by customers which usually falls below their expectations [49]. Generally, service failures result from employees' mistake, problem, or error while delivering service to customers [50]. Customers tend to remember service flaws more readily than great service [51]; thus, high-quality service implies few incidents of service flaws [52]. However, it is not possible to ensure completely error-free service because of the unique nature of service which is co-created with customer participation [53,54].

The most distinctive trait of the service industry compared with the manufacturing industry would result from customers' participation in the production process [54]. Customers are not separated from the service provision process in the restaurant setting, thus, they experience service failures in various stages of their dining experience. Several studies divided service failures into two stages: processes and outcomes [55,56]. Accordingly, service failures began to be addressed with respect to these two dimensions [57,58]. Process failures describe an error occurring in how the service is provided while outcome failures reflect an error in what is provided [59]. More specifically, process failures refer to a defect in delivering the core service and products (e.g., slow service, or rude or careless employees), representing the intangible elements of service delivery. For example, a long wait to be served or not being respected by servers are two common process failures [57,60]. Outcome failures refer to the failure of providing tangible products/service or core/basic service (e.g., overbooking, a bug in the food, overcooked food, or out of stock items). They mainly reflect tangible outcomes and major needs. Any problems with food, such as poor food quality or out-of-stock menu items are examples of outcome failures. Thus, customers experience service failures when they do not get what they pay for either process or outcome-based or both [61]. Thus, it is necessary to explore how different types of service failures have a more significantly negative impact on customers.

Numerous studies have addressed service failures and their negative effects on the business in a variety of contexts [50,62]. Negative consequences of service failures are reported, e.g., dissatisfaction, a decline in customer confidence; negative WOM behavior; customer defection; increased costs; loss of revenue; decreased in employee morale and performance [63]. Some dissatisfied consumers are provoked to engage in negative WOM while others are more likely to directly complain to the service provider [64,65]. WOM refers to verbal communication between people generally in the non-commercial context to exchange their thoughts, comments, or ideas [66]. WOM can be used to share verbally individual consumption experience either positive or negative in terms of product quality and service providers. An earlier study by Katz and Lazarfeld [67] found that WOM has a more powerful impact on consumers' purchase decisions than commercial advertising. Scholars asserted that if a service failure is not managed correctly, it leads customers to engage in negative WOM more actively [68,69]. Resnik and Harmon [70] stated that a complaint represents a consumer's negative emotional response towards unsatisfactory products/service quality. Singh [71] defined "consumer complain behavior" as a set of behavioral or non-behavioral responses triggered by consumer dissatisfaction about their purchase experience. Scholars asserted that a complaint can be the most critical responses made by consumers because it negatively affects enterprises including restaurants, thus it has received attention from academics and practitioners [72,73]. Lantos [74] identified different types of consumer complaint behavior such as voice responses (i.e., directly complaining to a service provider), private responses (i.e., sharing their experience within

private circles such as friends and family member) and third-party responses (complaining to an external third party but are not directly involved in discontented exchanges). Customer complaints are inevitable when service failures occur, and service failures increase customer complaints in future failed encounters [75].

Scholars have attempted to better understand the motivations beyond consumers' responses to service failures. The literature identified such motivations including personality traits [76], attitudes toward complaining [77], personal values [78], and so on. In addition, scholars strived to better understand reasons why customers make different responses towards their unpleasant service failure experiences [79]. For example, consumers who are more likely to complain they expect value of obtaining redress, availability of direct compensation, ease, and convenience of obtaining redress. In contrast, consumers who are less likely to complain think that their complaints may be of small value or may receive no attention. Thus, they would consider switching brands which is much easier than complaining. As another reason for not complaining, they do not know how to complain, or they may suffer from long procedure of bureaucracy, leading them to spread negative WOM [80]. In conclusion, consumer behaviors due to service failures can vary based on their own motivations.

### 2.3. Effects of Negative WOM on Switching Intention and Revision Intention

Literature provided clear evidence that WOM enabling consumers to share information and evaluations about their purchase experience guides and directs potential buyers in their choice of products or services [81,82]. Consumers feel more comfortable sharing their experiences, ratings, or knowledge about certain products and services [83]. It became common for consumers to rely on WOM before they make a purchasing decision to reduce risks and uncertainties [84,85]. In consideration of this powerful impact that WOM has on consumer purchase decisions, organizations begin to seek negative WOM that describes consumers sharing experiences about service failures and at the same time, asking for others' understanding and empathy [86,87].

Consumer switching is one of the main concerns of service providers [88], because negative WOM substantially impacts consumers' attitudes and influences switching behavior [89]. Switching intention represents the extent to which consumers are willing to switch from one company to another company [90]. Kuruuzum and Koksal [91] suggested customers' WOM represents their future behavior. Thus, negative WOM leads customers to develop switching intentions to other brands or businesses and to reduce revisit intention [92,93]. Following the discussion, this study proposed that negative WOM about restaurant service failures significantly boosts consumer switching intention, however, reduces their revisit intension. The following hypotheses are developed as follows:

**Hypothesis 1 (H1).** *Negative WOM about restaurant service failures significantly increases consumer switching intention.*

**Hypothesis 2 (H2).** *Negative WOM about restaurant service failures significantly reduces consumer revisit intention.*

### 2.4. Effects of Direct Complaints on Switching Intention and Revision Intention

Customer complaints are genuine actions of expressing their negative experiences about products and services, leading to a high possibility of replacing the current service provider with others [94]. Switching intention refers to a customer's intention to voluntarily and involuntarily end a relationship with a company [95]. These switching behaviors are stated as the last means of dissatisfied customers' complaining behavior [96]. Lai et al. [97] found that complaints significantly influence revisit intention. Chan et al. [15] suggested that if the restaurant customers' complaints are not resolved, these customers will not return to the restaurant. Literature on restaurant service approached revisit intentions as an affirmed likelihood to revisit a certain restaurant with positive towards both the service

provider and the restaurant [98,99]. Thus, restaurant guests who have experienced low service quality and direct complaints are more likely to have low revisit intention. Given this, the following hypotheses are proposed:

**Hypothesis 3 (H3).** *Direct complaints about restaurant service failures have a positive effect on consumer switching intention.*

**Hypothesis 4 (H4).** *Direct complaints about restaurant service failures have a negative effect on consumer revisit intention.*

*2.5. Moderating Effects of Guanxi within Restaurant Service Failures*

*Guanxi* has extensive influence in business conditions thus, it is regarded as being essential for business survival and performance growth [100]. Hence, most prior studies on *Guanxi* have been carried out from a perspective of relationship marketing and knowledge sharing and negotiation management in both B2B and B2C settings [101–103] for tourism [104] and hospitality industry [105]. Liu and McClure [106] asserted that consumer reactions towards unpleasant service experiences can vary according to cultural difference, because consumers who have exposed to different cultures in their daily life experiences tend to have different types of complaint behavior and intentions. For instance, collectivist consumers such as Asian consumers are less likely to complain because they consider that it may make them lose face, leading them to express their responses in private compared with those in individualistic cultures such as western countries. Based on traditional Chinese culture, expressing anger, or using radical voices in public is considered highly impolite and childish behavior, further damaging self-image.

An early study conducted by Leung et al. [29] reported that *Guanxi* results in well-developed information exchanges among Chinese. Literature asserted that Chinese customers tend to spread word-of-mouth because they tend to avoid reactions that jeopardize the interest of employees connected through *Guanxi* [59,107]. In contrast, directly complaining about service failures are less seen among the Chinese because they would feel like they lose face [108,109] Mittal et al. [110] supported this, demonstrating that customers who have built a strong connection with a service provider are less likely to complain about their service failures because they are worried about disrupting their relationship. In the Chinese business context (including restaurants), employees and customers are connected through a personal network of *Guanxi*. Customers become more trusting when they perceive special efforts made by the business or employees to take care of their needs and solve problems. *Guanxi* leads consumers to seek private and direct complaint channels and express their dissatisfaction directly to a service provider [59].

Well-established *Guanxi* between consumers and employees can develop great customer loyalty to service providers, further reducing any negative reactions to service failures [43,59]. Li and Liu [59] demonstrated that *Guanxi* makes hotel guests more tolerant about service failures and reduces the possibility of terminating transaction relationships. This implies that when Chinese consumers are directly complaining about service failures to a service provider, they may expect service failure recovery with high intent to stay at the business rather than switching to other restaurants. Thus, this study expects that when *Guanxi* is well-developed between customers and restaurant employees may have significant moderating roles on the relationships between negative WOM, direct complaints, switching intention, and revisit intention (see Figure 1). Thus, the following hypotheses are proposed:

**Hypotheses 5a and 5b.** *Guanxi moderates the effects of negative WOM on consumer switching intention (H5a) and revisit intention (H5b).*

**Hypotheses 6a and 6b.** *Guanxi moderates the effects of direct complaints on consumer switching intention (H6a) and revisit intention (H6b).*

**Restaurant Service Failures (Processes and Outcomes)**

**Figure 1.** Research model.

## 3. Methodology

### 3.1. Scenario Development

To achieve this study's objectives, two types of service failures (processes and outcomes) were developed based on the relevant literature on service failures using a scenario-based method [15,16,57]. The process failure scenario represents conditions related to service process failures which possibly lead to customers' negative emotions in a restaurant setting (i.e., inattentive service provides, a server's rude/discourteous behaviors, slow service, and disordered food delivery). The outcome failure scenario includes situations related to service performance failures which potentially lead to customers' dissatisfaction such as improperly cooked food and unavailable menu items. A pilot test was conducted with 200 Chinese diners using a convenience sample method to verify the effectiveness of service failures manipulations using a total of six items. Three items were used for the manipulation check of the process failures scenario. One example statement read "the server's attitude and service process at this restaurant were poor" for the process failures scenario. The other three items were used for the outcome failures scenario. One example item reads "the availability of food and properly cooked food in this restaurant was not proper." Based on the pilot test's results, the scenarios were revised for better readability and clarity. The final scenarios are presented in Table 1.

### 3.2. Measurement Instruments

Through a comprehensive literature review process, the study' survey instrument was developed by including several sections (see Table 2). The first section was established to measure "negative WOM", "direct complaints", "switching intention", and "revisit intention". More specifically, "negative WOM" was measured using five items adapted from East et al. [111]; "direct complaints" were evaluated using three items based on Blodgett et al. [96]; "switching intention" was assessed using three items from Shin and Kim [112]; and "revisit intention" was measured using three items from Blodgett et al. [96]. The second section was developed to assess a degree of *Guanxi* that respondents possess towards restaurant employees and owner. A total of five items adopted from Yen et al. [43] were used to measure *Guanxi.* The last section included items asking general information

of respondents regarding socio-demographic characteristics (i.e., gender, age, marital status, numbers of family members, education background, and monthly income) and their dining-out patterns.

**Table 1.** Process and outcome failures scenarios.

| Process failures: |
| --- |
| "You arrive at the restaurant to dine out. You are seated at a table, but you notice that the table has not yet been cleaned. You call the server and request they clean your assigned table. However, the server ignores you, and avoids making eye contact with you. Instead of responding to your request, the server chats with another coworker. Finally, the server arrives at your table and cleans it. You notice their uniform is not clean. The server then proceeds to take your order. You notice the server is not wearing a face mask. The server offers no apology or explanation." |
| **Outcome failures:** |
| "You arrive at the restaurant to dine out. The server comes to your table and takes your order. The server informs you that your selection is no longer available. You select another menu item. When your order arrives, you immediately notice it has been improperly prepared. You express your dissatisfaction to the server, who offers no apology or explanation." |

**Table 2.** Measurement instruments.

| Negative WOM (East et al., 2007) |
| --- |
| N1: If asked about this restaurant, I would respond negatively<br>N2: I would express my negative experiences about this particular restaurant's service provider if it were to be mentioned in conversation with my friends or relatives.<br>N3: I would report all my negative service experiences to my friends or relatives. |
| **Direct complaints (Bodgett et al., 1997)** |
| D1: I will forget the unsatisfactory experience and not complain to anyone (R)<br>D2: I will complain to the restaurant employee immediately upon experiencing dissatisfaction.<br>D3: will demand the restaurant's employee take proper action to correct the situation immediately after experiencing dissatisfaction. |
| **Switching intention (Shin and Kim 2008)** |
| S1: Due to my personal experience with service failures issues, I intend to no longer revisit this restaurant.<br>S2: Based upon this restaurant service failures experience, I will no longer continue a relationship with this employee.<br>S3: I will no longer patronize this restaurant based upon this service failures experience. |
| **Revisit intention (Blodgett et al., 1997)** |
| R1: I intend to return this restaurant to dine again.<br>R2: Compared with other restaurants, I consider this particular one to be my top dining choice.<br>R3: I remain committed to visiting this restaurant again. |
| *Guanxi* (Yen et al., 2011) |
| G1: I and an employee at this restaurant as friends are able to enjoy relaxed, comfortable conversation with one another.<br>G2: I feel a sense of obligation to at least one restaurant employee.<br>G3: Being able to "give and take" represents an important part of the relationship between myself as a restaurant customer, and an employee at this restaurant.<br>G4: I am willing to provide a favor for an employee when he/she requests one at this restaurant. |
| G5: Employees at this restaurant are concerned with my needs. |

The present study used scenario-based service failures in a restaurant setting which were adapted from previous relevant studies [15,16,57]. Respondents were first asked to answer whether they have experienced service failures and have *Guanxi* with one of its restaurant's employees. In doing so, respondents indicated their negative feelings and

levels of dissatisfaction regarding those service failures situations. Scenario reality and manipulation check were conducted to validate the appropriateness of the study's scenarios by following Chan et al. [15]. Then, all items used in this study were rated on a 7-point Likert scale (1 = strongly disagree to 7 = strongly agree).

### 3.3. Data Collection and Sampling

Our survey questionnaire was launched on one of the most frequently used Chinese online survey websites "sojump.com," which has been well-known as a reliable online survey website widely employed by Chinese researchers conducting a scenario experiment study [59]. Adopting a convenience sampling method, the survey questionnaire was distributed through WeChat, which is a do-everything social network that is owned by China's Tencent, after reading the scenarios. Participants were randomly assigned to one of the two scenarios (either process failures or outcome failures) and asked for imaging that they visited a restaurant for dinner. After reading one scenario, participants were asked to indicate their opinions about the service failures situations in consideration of their relationships developed based on *Guanxi* with restaurant employees or owners. As a result, a total of 439 (211 for the process failures and 228 for the outcome failures) responses were obtained and used for the study's analysis during a two-week period (1–15 September 2020).

All respondents reported their gender, age, marital status, companion, spending, occupation, education, and income which seemed to be no significant different in the characteristics between Study 1 (process failures) and Study 2 (outcome failures). A total of 211 valid responses were collected for Study 1 (process failures). Of the participants, 68.7% were male and 31.3% were female. There were slightly greater unmarried (46%) than those of the married (54%). The participants were 25–34 years old (42.7%) and held a bachelor's degree (39.8%) or above (17.1%). The majority of the participants were dining with friends (63.5%), relatives (25.1%), and colleagues (11.4%). Their monthly income represented to be ¥5000–¥10,000 (34.6%), follow by ¥3000–¥5000 (32.7%), ¥10,000–¥20,000 (14.2%), less than ¥3000 (13.8%) and ¥20,000 and above (4.7%)

In addition, a total of 228 valid responses were collected for Study 2 (outcome failures). Of the participants, 59.6% were male and 40.4% were female. There were slightly greater unmarried (56.6%) than married (43.4%). The participants were 25–34 years old (45.2%) and held a bachelor's degree (50%) or above (16.2%). The majority of the participants were dining with friends (61.4%), relatives (29.4%), colleagues (9.2%), and their monthly income represented to be ¥5000–¥10,000 (35.5%), followed by ¥3000–¥5000 (26.7%), less than ¥3000 (21.9%), ¥10,000–¥20,000 (11%), and ¥20,000 and above (4.8%).

## 4. Results

### 4.1. Reliability and Validity of the Measures

Confirmatory factor analysis was conducted to estimate reliability and validity of the measures used for this study. Among the three items of direct complaints, the factor loading of one item (D1: I will forget the unsatisfactory experience and not complain to anyone) was found to be smaller than 0.5; thus, it was eliminated. CFA was then re-run and its results are presented in Table 3. The assessment of the measurement model in this study indicates that it is an acceptable model fit ($\chi^2$ = 198.30; $df$ = 94; $\chi^2/df$ = 2.11; CFI = 0.93; GFI = 0.90; NFI = 0.88; RMSEA = 0.06). All standardized factor loadings exceeded 0.50 at a significance of $p < 0.001$. Cronbach's alpha coefficients of each construct ranging between 0.740 and 0.853 which are all above the reference value of 0.5 (Nunnally 1978). All average variance extracted (AVE) values were above 0.500 (ranging from 0.500 to 0.660). All composite reliability values exceeded the threshold value of 0.70 (ranging from 0.707 to 0.812). Thus, convergence validity of this study measures was supported [113].

**Table 3.** Results of confirmatory factor analysis.

| Constructs | Factor Loadings | CCR | AVE | Cronbach's $\alpha$ |
|---|---|---|---|---|
| Negative WOM | | 0.812 | 0.659 | 0.852 |
| N1 | 0.833 | | | |
| N2 | 0.867 | | | |
| N3 | 0.729 | | | |
| Direct complaints | | 0.740 | 0.547 | 0.740 |
| D2 | 0.644 | | | |
| D3 | 0.824 | | | |
| Switching intention | | 0.812 | 0.660 | 0.853 |
| S1 | 0.850 | | | |
| S2 | 0.731 | | | |
| S3 | 0.850 | | | |
| Revisit intention | | 0.729 | 0.531 | 0.772 |
| R1 | 0.658 | | | |
| R2 | 0.686 | | | |
| R3 | 0.897 | | | |
| *Guanxi* | | 0.707 | 0.500 | 0.832 |
| G1 | 0.760 | | | |
| G2 | 0.758 | | | |
| G3 | 0.652 | | | |
| G4 | 0.722 | | | |
| G5 | 0.622 | | | |

Notes: $\chi^2 = 216.154$, $df = 94$, $\chi^2/df = 2.30$; CFI = 0.93, NFI = 0.88, GFI = 0.90, RMSEA = 0.06; CCR = 'composite construct reliability,' AVE = 'average variance extracted.'

Table 4 shows the means, standard deviations, and correlation coefficients of the study constructs. Correlation analysis found that "negative WOM" was positively related to "switching intention" and also negatively associated with "revisit intention." "Direct complaints" was positively related to "switching intention"; however, was negatively associated with "revisit intention." *Guanxi* was positively related to "direct complaints" and "revisit intention"; however, it was negatively associated with "negative WOM" and "switching intention." All values of the square root of the average variance extracted (AVE) were larger than the correlation coefficients among the constructs, which supported discriminant validity of the measures [113].

**Table 4.** Discriminant validity and correlations.

| | Mean | S.D | NWOM | DCOM | SI | RI | GX |
|---|---|---|---|---|---|---|---|
| NWOM | 4.58 | 1.39 | 0.812 [a] | | | | |
| DCOM | 4.59 | 1.38 | 0.359 | 0.740 | | | |
| SI | 4.38 | 1.44 | 0.810 | 0.428 | 0.812 | | |
| RI | 3.58 | 1.27 | −0.484 | −0.016 | −0.467 | 0.729 | |
| GX | 4.95 | 1.22 | −0.100 | 0.346 | −0.076 | 0.236 | 0.707 |

Notes: NWOM = "negative word of mouth," DCOM = "direct complaints," SI = "switching intention," RI = "revisit intention"; [a] Diagonals: square root of AVE from the observed variables by the latent variables. AVE = 'average variance extracted.'

### 4.2. Results of Testing Hypotheses 1 to 4

To test the hypothesized relationships in this study, a structural equation model was developed. In addition, respondents' age, gender, marital status, and income were included into the SEM to control their potential effects on the relationships. First, we tested hypotheses $1_{p \text{ (process failures)}}$ through $4_{p \text{ (process failures)}}$, expecting the significant relationships between negative WOM, direct complaints, switching intention and revisit intention

within the process failures setting. As presented in Table 5, our structural equation model (SEM) was found to be acceptable with goodness-of-fit-indexes ($\chi^2/df$ = 2.15, CFI = 0.93; NFI = 0.90; GFI = 0.92). Regarding the relationships between "negative WOM," "switching intention," and "revisit intention," results revealed that "negative WOM" increased "switching intention" ($\beta$ = 0.782, $p$ < 0.001), but decreased "revisit intention" ($\beta$ = −0.422, $p$ < 0.001). Thus, hypotheses 1 and 2 were supported. Furthermore, our results found that "direct complaints" had no significant effect on "switching intention" ($\beta$ = 0.055, $p$ > 0.05). However, "direct complaints" significantly and positively influenced "revisit intention" ($\beta$ = 0.153, $p$ < 0.05). Since we expected the negative relationship between "direct complaints" and "revisit intention," Hypotheses 3 and 4 were not supported.

**Table 5.** Results of testing hypotheses $1_p$ through $4_p$ (process failures).

|  |  | Estimates | S.E. | C.R. | $p$ | Results |
|---|---|---|---|---|---|---|
| $H1_p$ | NWOM–SI | 0.782 | 0.080 | 9.809 | 0.000 *** | Supported |
| $H2_p$ | NWOM–RI | −0.422 | 0.071 | −5.935 | 0.000 *** | Supported |
| $H3_p$ | DCOM–SI | 0.055 | 0.090 | 0.609 | 0.543 | Not supported |
| $H4_p$ | DCOM–RI | 0.153 | 0.077 | 1.988 | 0.047 * | Not supported |

Notes: NWOM = 'negative word of mouth,' DCOM = 'direct complaints,' SI = 'switching intention,' RI = 'revisit intention'; *** $p$ < 0.001; * $p$ < 0.05; $\chi^2$ = 84.13, $df$ = 39, $\chi^2/df$ = 2.15; CFI = 0.93, NFI = 0.90, GFI = 0.92.

Second, hypotheses $1_{o\ (outcome\ failures)}$ through $4_{o\ (outcome\ failures)}$, expecting the significant relationships between negative WOM, direct complaints, switching intention and revisit intention within the outcome failures setting. As shown in Table 6, the goodness-of-fit-indexes of SEM were satisfactory ($\chi^2/df$ = 2.43, CFI = 0.93; NFI = 0.92; GFI = 0.93). We found the same effects of "negative WOM" on "switching intention" ($\beta$ = 0.718, $p$ < 0.001), and on "revisit intention" ($\beta$ = −0.227, $p$ < 0.01) as the results found in the process failures condition. Thus, hypotheses 1 and 2 were supported. In the same vein, we found that "direct complaints" had no significant effects on "switching intention" ($\beta$ = −0.009, $p$ > 0.05), but it had the significant and positive effect on "revisit intention." Hence, hypotheses 3 and 4 were not supported.

**Table 6.** Results of testing hypotheses $1_o$ through $4_o$ (outcome failures).

|  |  | Estimates | S.E. | C.R. | $p$ | Results |
|---|---|---|---|---|---|---|
| $H1_o$ | NWOM–SI | 0.718 | 0.091 | 7.891 | 0.000 *** | Supported |
| $H2_o$ | NWOM–RI | −0.227 | 0.086 | −2.648 | 0.008 ** | Supported |
| $H3_o$ | DCOM–SI | −0.009 | 0.082 | −0.106 | 0.916 | Not supported |
| $H4_o$ | DCOM–RI | 0.260 | 0.098 | 2.665 | 0.008 ** | Not supported |

Notes: NWOM = "negative word of mouth," DCOM = "direct complaints," SI = "switching intention," RI = "revisit intention"; *** $p$ < 0.001, ** $p$ < 0.01; $\chi^2$ = 95.13, $df$ = 39, $\chi^2/df$ = 2.43; CFI = 0.93, NFI = 0.92, GFI = 0.93.

### 4.3. Results of Testing Hypotheses 5 and 6

First, Guanxi was tested to see if it has the moderating roles on the hypothesized relationships among the proposed constructs (i.e., negative WOM, direct complaints, switching intention, and revisit intention) within the process failures setting. Using the mean value of *Guanxi*, the respondents were divided into two groups: the high-*Guanxi* ($n$ = 105) and low-*Guanxi* ($n$ = 106) groups (see Table 7). Then in order to test if there are significant differences in the relationships between the study constructs between the high-*Guanxi* and the low-*Guanxi* group, a multi-group analysis was conducted.

**Table 7.** Results of testing H5$_p$ and H6$_p$ (process failures).

| | | High-*Guanxi* (*n* = 105) | | Low-*Guanxi* (*n* = 106) | | $\chi^2$ **Difference** | **Results** |
|---|---|---|---|---|---|---|---|
| | | **Coefficient** | **C.R** | **Coefficient** | **C.R** | | |
| H5$_{ap}$ | NWOM–SI | 0.799 | 6.355 *** | 0.714 | 8.100 *** | $\Delta\chi^2(1)$ = 0.507, $p$ > 0.05 | Not supported |
| H5$_{bp}$ | NWOM–RI | −0.589 | −4.531 *** | −0.606 | −3.496 *** | $\Delta\chi^2(1)$ = 1.036, $p$ > 0.05 | Not supported |
| H6$_{ap}$ | DCOM–SI | −0.051 | −0.514 | 0.228 | 2.561 ** | $\Delta\chi^2(1)$ = 4.463, $p$ < 0.05 | Supported |
| H6$_{bp}$ | DCOM–RI | −0.033 | −0.313 | 0.247 | 2.002 * | $\Delta\chi^2(1)$ = 0.808, $p$ > 0.05 | Not supported |

Notes: NWOM = "negative word of mouth," DCOM = "direct complaints," SI = "switching intention," RI = "revisit intention"; *** $p$ < 0.001, ** $p$ < 0.01, * $p$ < 0.05.

Results showed that the effect of "negative WOM" upon "switching intention" was found to be significantly positive in both the high-*Guanxi* group (β = 0.799, $p$ < 0.001) and in the low-*Guanxi* group (β = 0.714, $p$ < 0.001) under the process failures condition. The difference in the path coefficients was not significant ($\Delta\chi^2$ (1) = 0.507, $p$ > 0.05). Thus, hypothesis 5a$_p$ was not supported. Additionally, the effect of "negative WOM" upon "revisit intention" was found to be significantly negative in the high-*Guanxi* group (β = −0.589, $p$ < 0.001) and in the low-*Guanxi* group (β = −0.606, $p$ < 0.001), indicating no significant difference in the path coefficients ($\Delta\chi^2$ (1) = 1.036, $p$ > 0.05). Thus, hypothesis 5b$_p$ was not supported either.

Results found that the effect of "direct complaints" upon "switching intention" was negative but insignificant in the high-*Guanxi* group (β = −0.051, $p$ > 0.05). However, such the relationship was significantly positive in the low-*Guanxi* group (β = 0.228, $p$ < 0.01). This difference was significant ($\Delta\chi^2$ (1) = 4.463, $p$ < 0.05). Thus, hypothesis 6a$_p$ was supported. In addition, the effect of "direct complaints" upon "revisit intention" was found to be negative but insignificant in the high-*Guanxi* group (β = −0.033, $p$ > 0.05) while its relationship was positive and significant in the low-*Guanxi* group (β = 0.247, $p$ < 0.05). However, this difference was not significant ($\Delta\chi^2$ (1) = 0.808, $p$ > 0.05). Thus, hypothesis 6b$_p$ was not supported.

Second, using the same methods, respondents were divided into two groups: the high-*Guanxi* (*n* = 126) and low-*Guanxi* (*n* = 102) groups to test the moderating roles of *Guanxi* on the hypothesized relationships within the outcome failures setting (see Table 8).

**Table 8.** Results of testing H5$_o$ and H6$_o$ (outcome failures).

| | | High-*Guanxi* (n = 105) | | Low-*Guanxi* (n = 106) | | $\chi^2$ **Difference** | **Results** |
|---|---|---|---|---|---|---|---|
| | | **Coefficient** | **C.R** | **Coefficient** | **C.R** | | |
| H5$_{ao}$ | NWOM–SI | 0.637 | 4.735 *** | 0.821 | 6.054 *** | $\Delta\chi^2(1)$ = 6.197, $p$ < 0.05 | Supported |
| H5$_{bo}$ | NWOM–RI | −0.145 | −1.035 | −0.456 | −3.334 *** | $\Delta\chi^2(1)$ = 4.142, $p$ < 0.05 | Supported |
| H6$_{ao}$ | DCOM–SI | 0.107 | 0.816 | −0.098 | −0.884 | $\Delta\chi^2(1)$ = 2.193, $p$ > 0.05 | Not supported |
| H6$_{bo}$ | DCOM–RI | 0.093 | 0.608 | 0.447 | 2.945 ** | $\Delta\chi^2(1)$ = 1.371, $p$ > 0.05 | Not supported |

Notes: NWOM = "negative word of mouth," DCOM = "direct complaints," SI = "switching intention," RI = "revisit intention"; *** $p$ < 0.001, ** $p$ < 0.01.

The effect of "negative WOM" upon "switching intention" was found to be significantly positive in both the high-*Guanxi* group (β = 0.637, $p$ < 0.001) and the low-*Guanxi* group (β = 0.821, $p$ < 0.001). This difference was significant ($\Delta\chi^2$ (1) = 6.197, $p$ < 0.05). Thus, hypothesis 5a$_o$ was supported. Moreover, the effect of "negative WOM" upon "revisit intention" was negative but insignificant in the high-*Guanxi* group (β = −0.145, $p$ > 0.05) while the effect was significantly negative in the low-*Guanxi* group (β = −0.456, $p$ < 0.001). Accordingly, the difference was significant ($\Delta\chi^2$ (1) = 4.142, $p$ < 0.05). Thus, hypothesis 5b$_o$ was supported.

Results found the effect of "direct complaints" on "switching intention" was insignificant in either the high-*Guanxi* group (β = 0.107, p > 0.05) or the low-*Guanxi* group (β = −0.098, $p$ > 0.05). This difference was not significant ($\Delta\chi^2$ (1) = 2.193, $p$ > 0.05). Thus, hypothesis 6a$_o$ was not supported. Furthermore, the effect of "direct complaints" on "revisit intention" was found to be insignificant in the high-*Guanxi* group (β = 0.093, $p$ > 0.05)



but its effect was significant and positive in the low-*Guanxi* group ($\beta$ = 0.447, $p < 0.01$). However, this difference was not significant ($\Delta\chi^2$ (1) = 1.371, $p > 0.05$). Thus, hypothesis $6b_o$ was not supported.

## 5. Conclusions and Implications

### 5.1. Conclusions

Overall, this current study concluded the significant relationships between restaurant guests' responses and their future behavior in the service failure settings. In addition, *Guanxi* was proven as an important source of competitive advantage for restaurant operational management. More specifically, this study found that Chinese restaurant guests who encountered service failures and spread negative WOM are more likely to have high switching intention. The significant and positive relationship between negative WOM and switching intention was consistently found in both the process failures and the outcome failures setting. This relationship is in line with the previous relevant literature [114]. Additionally, negative WOM significantly reduced revisit intention. Such an impact was found to be stronger within the process failure setting ($\beta$ = −0.422 ***) than in the outcome failure setting ($\beta$ = −0.227 **). A previous study by Ko and Kim [115] clearly stated that negative WOM can be a strong indicator for consumer negative attitudes and behavioral intention towards the business. Thus, the study finding confirmed the strong effect of negative WOM on revisit intention when Chinese diners experience service process and outcome failures.

No significant relationship between direct complaints and switching intention was found and this result was consistent in both the process failures and the outcome failures setting. However, this study revealed the positive relationship between direct complaints and revisit intention within the restaurant service failures setting. Such an effect was found to be stronger in the outcome failures setting ($\beta$ = 0.260 **) than in the process failures setting ($\beta$ = 0.153 *). These results implied that Chinese guests who encounter service failures and directly complain to the service provider might hope that the restaurant corrects those issues and enhance their service process and production adequately. This result can be supported by Bijmolt et al. [116] demonstrating the positive relationship between consumer complaints about low service quality and their repurchase intention. The authors explained that consumers who complain about their dissatisfaction with product or service providers are more willing to contribute to improving the problems, so those companies do not make similar mistakes. Those consumers are more likely to come back and repurchase, ultimately become loyal. In other words, consumers who experience service failures and directly complain are more likely to have higher repurchase intention compared to consumers who do not complain. This can be again supported by a more recent study that demonstrated when customers directly complained to a service provider, their intentions to revisit increased in the hotel industry [117].

Regarding the results of testing the moderating roles of *Guanxi*, this study revealed that when process-related failures occur, the negative relationship between direct complaints and switching intention weakened in the high-*Guanxi* group than in the low-*Guanxi* group. When outcome-related failures occur, the positive effect of negative WOM on switching intention was weaker in the high-*Guanxi* group than in the low-*Guanxi* group. Additionally, the significantly negative relationship between negative WOM and revisit intention turned insignificant in the high-*Guanxi* group while such a relationship was still strongly negative in the low-*Guanxi* group. Xiucheng and Jianhua [118] demonstrated that *Guanxi* developed within Chinese society affected consumer reactions towards service failures. More specifically, *Guanxi* played a critical role in mitigating consumers' negative experiences of service failures on their post-consumption behavior. These findings suggested that Chinese restaurant guests who have developed high *Guanxi* with restaurant owners or employees are less likely to switch to other restaurants, instead, they are more willing to continue to visit the restaurant even if they have experienced some process or outcome service failures.

### 5.2. Implications

It is commonly stated that no matter how hard companies try, even the best service company cannot avoid service failures in the hospitality industry. This clearly indicates that service failures are inevitable. Although today's restaurant management attempts to provide the best quality products and services to customers in the same vein, service failures common and frequent. This spreads negative WOM very easily and quickly leading to many consumers' loss. An early study by Harsono [119] stated that a consumer who has experienced service failures usually shares their unpleasant experience with 10 to 20 other people. Again, the author reported that only about 4% of customers among those dissatisfied customers tend to complain directly to the company or employees. The remaining customers (96%) have encountered severe service failures, they neither complain nor spread negative WOM. This implies that most of those customers are a huge hidden risk group because they would switch to other service providers. Likewise, service failures should be marked as a critical issue in the service industry, specifically for the restaurant business. With this in mind, this study particularly focused on Chinese guests and their culture-based personal interactions representing *Guanxi* to investigate its role in responses to restaurant service failures.

China's restaurant industry and its competition have been more intense [120]. Due to service failures, customer loss frequently occurs in restaurant settings, leading to high financial losses. Therefore, it is necessary to understand Chinese guests' attitudes and behavior towards service failures in the restaurant industry. With this in mind, the study was conducted to provide valuable information that can be applied to the development of more effective operational strategies for the sustained restaurant business. This study revealed that Chinese consumers are more concerned about process failures than outcome failures. This result implied that restaurant managers should develop well-organized employee selection, training, and reward programs. Moreover, to build *Guanxi* with guests, employees have to obtain a better culture-oriented perception of service failure's severity from a Chinese cultural perspective. Additionally, in terms of negative impacts of outcome failures on customers' behavioral intentions, restaurant management should develop regular checklists regarding cooking and service methods so that restaurant employees follow a code of conduct that empowers them to provide standardized menu quality items and services.

This study confirmed that *Guanxi* plays a substantial role in a service failure setting, providing clear evidence that restaurants need to utilize *Guanxi*'s principles to benefit themselves by developing more effective consumer relationship marketing. Chinese cultural-based personal relationships developed within the customer-service provider can determine consumer loyalty. Thus, this study provides the following suggestions to build *Guanxi* with restaurant guests. First, restaurant employees need to improve their clear and effective communication skills. When they make some errors in a service provision process, they have to provide clear explanations about that issue to consumers and ask them for their understanding and forgiveness. Second, restaurant employees should strive to keep their relationship close and long-term. It is not easy to form customer loyalty to a specific business in a short time. Thus, restaurants need to develop relationship marketing programs that enable long-term customer retention. Lastly, restaurant employees should be trained to learn instantly about what their consumers need and want and also to give rapid feedback about any issues before customers develop any intentions to switch or negative WOM.

### 5.3. Limitations and Recommendations for Future Studies

This study collected data from only Chinese guests. However, there would be significant cultural differences in customer responses towards service failures between Chinese and others. Thus, future studies need to obtain more rigorous data from many different territorial areas and investigate if there would be some significant differences from our findings. This study found no significant effect of respondents' demographic characteristics

on their behavioral intentions towards restaurant service failures. However, there could possibly be significant variance in response to restaurant service failures depending on customer demographics. Thus, future studies need to obtain more rigorous data and include those factors in their research framework. It is also possible that there would be additional vital factors that affect customer responses to service failures. Therefore, future studies need to identify other factors and investigate how they affect the consumer service failure experiences-response model.

**Author Contributions:** Conceptualization, C.Z., J.P. and M.C.; Methodology, C.Z., J.P., M.A.B. and M.C.; Validation, C.Z. and M.C.; Formal Analysis, C.Z., J.P. and M.C.; Investigation, C.Z., J.P. and M.C.; Resources, C.Z. and J.P.; Data Curation, C.Z., J.P. and M.A.B.; Writing—Original Draft preparation, C.Z. and J.P. and M.C.; Writing—Review and Editing, M.A.B. and M.C.; Visualization., C.Z., J.P. and M.C.; Supervision, M.A.B. and M.C.; Project Administration, M.A.B. and M.C. All authors have read and agreed to the published version of the manuscript.

**Funding:** This research received no external funding.

**Institutional Review Board Statement:** Ethical review and approval were waived for this study. Although it was a human subject study, the research design did not involve ethical issues and included adequate provisions to maintain the privacy interest of participants.

**Informed Consent Statement:** Informed consent was obtained from all subjects involved in the study.

**Data Availability Statement:** Data sharing not applicable. The data are not publicly available due to participants privacy.

**Conflicts of Interest:** The authors declare no conflict of interest.

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
