# Peer review of "Understanding Customer Responses to Service Failures during the COVID-19 Pandemic for Sustained Restaurant Businesses: Focusing on Guanxi"

_sustainability, doi:10.3390/su13063581_

Round 1

Reviewer 1 Report

I thank the authors for the work they have invested thus far in this study. Overall I was challenged with this article on a number of fronts. 

A core issue is that the authors have not attempted to explicitly connect their study with the aims and scope of Sustainability. In terms of subject area, the authors need to be clear how their work on consumer behaviors flowing from perceived service failures as moderated by cultural norms of familiarity obligation align with the goals of this particular journal. 

The writing style, as presented, used a lot of descriptive academic-style verbiage which, unfortunately, does not provide the reader with a clear idea of what is being examined or why. I would state that the authors appear to have a clear intention and direction with their study. However, it is not coming across clearly to their audience and I was left confused in numerous areas after close reading multiple times.

Abstract:

The concept of "Guanxi" is surfaced without any attempt to infer a meaning in the abstract. Therefore, it is immediately difficult to understand the direction of this work. The abstract should be written in clear terms that bring in the reader; the body of the work is a better location to present and explore the meaning of "Guanxi". 

The phrase "so as to formulate effective target restaurant market strategies" is unclear, yet also reinforces the above comment about fit. It appears at this point the article is about business strategies and customer decisions; the connection to sustainable subject areas does not flow from this. 

Recommend that the abstract be rewritten with clarity at the forefront. What is the research question of this piece? How does it connect to a topic within the realm of sustainability? What are the core constructs involved?

Introduction:

There are numerous English verb and phrase issues here which should be captured during the review phase.

The authors have presented a large list of statements from the literature reviewed. As a reader, I am left asking two questions. Firstly, why has this literature been selected and why is it being presented? This is a question of context. The literature should work with the situation that the authors are trying to explicitly establish as important; instead, the data of the foodservice industry in China is presented to speak for itself. Second, how has the literature been critically evaluated and applied for the use of justifying the value of this research? This is a question of application.

A catering market is not the same as dine-out restaurants. (Catering is foodservice provided at remote sites, such as hotels, events, even airlines, where food is not made available by the site operator.)

What it meant by ‘frequently’ dined out in a week? (2 times? 5 times?)

The Chinese are well-known as the customers who dine out frequently? (well known by whom?)

The operationalization of ‘Guanxi’ remains unclear. The authors seem to have an understanding that is being applied, but the definition of the term, and therefore its value to the work, remains cloudy and forces the reader to piece together an understanding. The authors do attempt to provide a better definition on pg 3 (‘mutual empathetic understanding, sharing of feelings and emotional identification’), but still have yet to clearly unpack this term. As it becomes a moderating variable within the model, it is critical to provide a clear meaning.

Recommend that the introduction be reorganized to provide a clear flow into the study to the reader. What are the key variables? Where is the gap? Why is further study around these constructs and/or within this cultural context valuable? It would also be valuable to clearly outline the methodology – you will be presenting a sample population with two service failure scenarios (process or content), accessing if they would directly complain, use negative WOM (or both), exposing the impacts on these two options with intentions to switch or revisit, then looking at if Guanxi moderates outcomes.

Methodology:

Why apply two different types of service failure? The model now has two various service failures, two potential behaviors (in any combination), two potential outcomes (in any combination), and a single moderator on every flow path. With so many moving parts, it is critical that the model is more clearly explained.

On pg 7, it reads, “service failures were developed based on respondents’ real personal relationships with restaurants owners or employees and scenario-based experiences”. It is unclear what this means. Recommend that you revise to state what you did in a clear, concise manner.

Why was a pilot study done of 200 diners for a study that only collected 439 valid responses? How was the pilot study conducted – also through panel data?  

Who was the target population for this study? Why collect and analysis demographic data when this was not part of the proposed analysis? It is stated that on pg 8 that there seemed to be no significant differences between study 1 and study 2 populations – was this tested for significant differences in response to a selection of questions? This is important as there is a large enough variance in areas such as gender, marital status, etc…that responses may (or may not) that these factors might need to be accounted for.

Overall:

I believe the authors have a clear intention in their work which is not yet being fully / clearly presented in their written submission. I would suggest that the comments around clarity in terms, in design, and in processes be addressed regardless of the home this piece is targeted towards. Should the authors by invited to resubmit their work to Sustainability, it is of utmost importance that they somehow ground this paper with one of the core subject areas of the journal and/or within the recent literature and published work of the journal (thereby also creating a clear and obvious fit with the journal).

Reviewer 2 Report

 The topic of the paper is interesting, but serious concerns must be taken into account. A research paper's abstract should contain the problem, the study's purpose, methods, data analysis, results, and conclusion. It should be restructured. 

Objective/Gap: the need for this research is not clearly stated. The main objectives of the study are not clear. The main research questions are not well developed. The introduction should start with a discussion of the scope and significance of the issue and or problem. Next, the manuscript needs a review of the literature that should provide the reader with a synthesis of previous work.

Heading 2.4: Effects of Direct Complaints on Switching Intention and Revision Intention (Line 204) should be supported by recent studies/literature.

Theoretical development: The paper did not discuss any theory to support arrangements. The theoretical assumptions of the research question development are confusing and are not well justified.

The methodology section is well explained.

The conclusion and discussion should be mentioned separately.

Future studies and study limitations should be under the sub-heading of discussion or conclusion.

The paper should be improved substantively before considering a publication. First, as an empirical study, it fails to follow the standard process. Second, you'd better discuss the theoretical gap in the introduction section and make your study based on solid theoretical foundations.

Round 2

Reviewer 2 Report

The manuscript can be published after proofreading.